# Application of a Synthetic Microbial Community to Enhance Pepper Resistance Against *Phytophthora capsici*

**DOI:** 10.3390/plants14111625

**Published:** 2025-05-26

**Authors:** Tino Flory Bashizi, Min-Ji Kim, Kyeongmo Lim, GyuDae Lee, Setu Bazie Tagele, Jae-Ho Shin

**Affiliations:** 1Department of Applied Biosciences, Kyungpook National University, Daegu 41566, Republic of Korea; tino.bashizi@knu.ac.kr (T.F.B.); lkm3519@knu.ac.kr (K.L.); leegyuedae@knu.ac.kr (G.L.); 2NGS Core Facility, Kyungpook National University, Daegu 41566, Republic of Korea; tbd01188@knu.ac.kr; 3Department of Microbiology and Plant Pathology, University of California Riverside, Riverside, CA 92507, USA; setu.tagele@ucr.edu

**Keywords:** *Capsicum annuum*, plant growth-promoting rhizobacteria, rhizosphere microbiome, sustainable agriculture, SynCom

## Abstract

Pepper (*Capsicum annuum*) production faces significant challenges from soil-borne pathogens, particularly *Phytophthora capsici*, which induces root rot and damping-off diseases. Management of this pathogen remains challenging owing to the scarcity of resistant cultivars and the ineffectiveness of chemical control methods. A single strain has been used to prevent pathogenic disease, and this approach limits the exploration of consortia comprising different genera. In this study, we isolated five bacterial strains (*Bacillus* sp. T3, *Flavobacterium anhuiense* T4, *Cytobacillus firmus* T8, *Streptomyces roseicoloratus* T14, and *Pseudomonas frederiksbergensis* A6) from the rhizosphere of healthy pepper plants. We then applied this 5-isolate synthetic microbial community (SynCom) to *Capsicum annuum* to evaluate its efficacy in improving pepper resilience against *P. capsici*. The SynCom members exhibited phosphate solubilization, indole-3-acetic acid production, catalase activity, siderophore synthesis, and strong antagonism against *P. capsici*. The SynCom reduced disease severity and enhanced the growth of pepper plants. Furthermore, the beneficial genera such as *Bacillus*, *Fusicolla*, and *Trichoderma*, significantly increased in the rhizosphere of pepper after the application of the SynCom. Microbial functional prediction analysis revealed that these microbial shifts were associated with nitrogen cycling and pathogen suppression. Our SynCom approach demonstrates the effectiveness of microbial consortia in promoting the growth of pathogen-infected plants by reprogramming the microbial community in the rhizosphere.

## 1. Introduction

To meet the demands of a growing global population, projected to reach 9.7 billion by 2050, agricultural productivity must be increased sustainably. This challenge calls for developing sustainable practices that minimize environmental impacts such as soil degradation, groundwater contamination, and biodiversity loss [1]. Traditional dependence on chemical fertilizers and pesticides, although effective in the short term, presents several disadvantages. These approaches contribute to environmental pollution, promote the emergence of resistant pests and pathogens, and disrupt beneficial soil microbial communities, ultimately reducing soil fertility and causing ecosystem imbalances. Conversely, plant growth-promoting rhizobacteria (PGPR) offer a promising and eco-friendly alternative by enhancing plant growth and resilience without the adverse environmental effects associated with conventional methods [2].

PGPR demonstrates significant potential in improving plant health via nutrient solubilization, phytohormone synthesis, and disease suppression, especially within the rhizosphere, where these microbes interact with plant roots. In addition to these functions, PGPR contribute to plant nutrition through phosphate solubilization, nitrogen enrichment via biological fixation or ammonia production, and siderophore-mediated iron acquisition, ultimately improving root development and physiological performance [3,4]. Furthermore, synergistic interactions between bacterial consortia and arbuscular mycorrhizal fungi have been shown to enhance nutrient availability and disease tolerance, even in nutrient-poor soils [5]. Specifically, PGPR suppress pathogens through mechanisms such as competitive exclusion, production of antimicrobial compounds, and induction of systemic resistance [6]. However, despite their promise, effectively harnessing these bacteria remains challenging. Previous studies show that while individual strains of PGPR can promote plant growth, their effectiveness in suppressing diseases is often inconsistent owing to the complex interactions within the rhizosphere [7]. Moreover, single-strain inoculants face limitations in sustaining long-term benefits and adapting to diverse environmental conditions [8]. Therefore, developing a synthetic microbial consortium (SynCom) may offer a more robust solution. By combining multiple strains, SynCom can complement individual weaknesses, offering better resilience and broader functionality [9]. However, optimizing SynCom requires overcoming challenges such as microbial antagonism or instability in consortial interactions under field conditions.

Among plant pathogens, *Phytophthora capsici* poses a significant threat to pepper (*Capsicum annuum* L.) production, causing severe crop losses under favourable conditions, leading to economic impacts estimated at billions of dollars annually in global pepper production. These losses disproportionately affect tropical, subtropical and temperate regions, where pepper is a staple crop and environmental conditions often favor the pathogen [10]. Environmental factors such as waterlogged soils and high humidity exacerbate outbreaks of this aggressive pathogen, whose rapid adaptation and survival mechanisms make it particularly challenging to manage. The inoculum typically originates from contaminated soils, infected plant residues, irrigation water, or previously infected transplants, allowing the pathogen to persist and initiate infection under favorable conditions. This pathogen also affects crops like tomatoes, cucumbers, and melons, highlighting its broad significance in agricultural systems worldwide. Its prevalence varies based on climatic conditions, crop types, and agricultural practices, with warm and humid regions being particularly vulnerable. Chemical fungicides often fail to provide sustainable control owing to their environmental effects and the emergence of resistant strains [11]. Biocontrol strategies have demonstrated the potential of microbial inoculants, especially actinomycetes such as *Streptomyces* species, which are renowned for their potent antifungal activity. Their efficacy is attributed to the production of diverse bioactive secondary metabolites, as exemplified by *Streptomyces rochei* IT20 and *S. vinaceusdrappus* SS14, both of which exhibited significant inhibitory effects against *P. capsici* despite their slower growth rates [12]. However, this slower proliferation limits their competitive establishment in the rhizosphere, revealing the need for integrating *Streptomyces* with fast-growing bacterial strains to enhance colonization efficiency and pathogen suppression [13]. Despite these promising insights, previous studies have focused on single-strain inoculants or lacked comprehensive evaluations of microbial consortia under conditions of high disease pressure, leaving critical gaps in understanding the full potential of synergistic microbial interactions [14].

Therefore, this study aims to address gaps in sustainable agriculture by developing a plant growth-promoting rhizobacteria (PGPR) consortium (SynCom) consisting of five bacterial strains—*Bacillus* sp. T3, *Flavobacterium anhuiense* T4, *Cytobacillus firmus* T8, *Streptomyces roseicoloratus* T14, and *Pseudomonas frederiksbergensis* A6—all isolated from the pepper rhizosphere. Pot experiment evaluated the combined effects of SynCom on plant growth promotion and disease suppression in pepper plants under high disease pressure induced by *Phytophthora capsici*. Building upon previous research, this study shifts the focus from single-strain applications to SynCom, addressing the paucity of studies investigating their role in plant health management. Moreover, it aims to address a key research gap by offering a controlled experimental framework to evaluate the effectiveness of consortia under disease-pressure conditions.

## 2. Results

### 2.1. Identification of Plant-Growth Promoting Rhizobacteria for SynCom Construction

Bacterial isolates were obtained from the rhizosphere soil of visually healthy pepper plants—defined as plants showing no symptoms of disease (e.g., wilting, necrosis, stunting), and exhibiting vigorous vegetative growth. These plants were in the early flowering stage but had not yet fruited at the time of sampling. Rhizosphere soil was serially diluted and plated on tryptic soy agar (TSA), and colonies were selected based on morphological differences. To assess biocontrol potential, individual isolates were screened for antagonistic activity against *P. capsici* KACC 40,157 using dual-culture assays on V8 agar plates. Each bacterial isolate was streaked 2 cm away from a central 5 mm plug of actively growing *P. capsici*, and inhibition zones were measured after 5 days of incubation at 25 °C. Experiments were performed in triplicate. Out of the screened isolates, five were selected to formulate a bacterial consortium based on superior biochemical traits: T14, which showed the strongest antagonistic activity; T3, with high phosphate solubilization; T4, with strong siderophore production; A6, with elevated IAA production and biofilm formation; and T8, effective in ammonia production. Although additional traits such as chitinase and lipase production were detected in some strains, the consortium composition prioritized functional complementarity for both pathogen suppression and plant growth promotion. Biofilm production and cellulase activity were confirmed on Congo red agar, where all selected strains produced halo zones indicative of extracellular matrix formation and cellulose degradation, respectively (Table 1). Strain identities were confirmed by 16S rRNA gene sequencing, and the sequences were deposited in GenBank under accession numbers listed in Appendix A.

The bacterial growth dynamics of the five strains—T3, T4, T8, T14, and A6—were monitored in tryptic soy broth (TSB) at 30 °C with shaking at 180 rpm. Growth curves were plotted based on colony forming units (CFU), and specific growth rates were calculated for each strain. As shown in Figure 1a, strains T4 and T8 reached the highest cell densities, followed by T3 and A6, while T14 exhibited the slowest growth rate among the five.

### 2.2. Changes in the Pepper Growth Through SynCom Application

The effect of SynCom on plant growth under pathogen-induced stress was evaluated using non-sterile soil (Figure 1b). When inoculated with the bacterial SynCom alongside the pathogen, the treatment group showed significant enhancements across all measured growth parameters compared to that of the pathogen-only control group (Figure 1c). Root length, plant height, fresh weight, dry weight, stem diameter, and chlorophyll content were significantly improved in the treated plants. Plant height increased significantly (*p* < 0.001), alongside fresh weight (*p* < 0.001) and dry weight (*p* < 0.01), indicating improved vigour and biomass accumulation. Root length also exhibited a significant increase (*p* < 0.01), suggesting enhanced rooting and nutrient uptake. Stem diameter was larger in treated plants (*p* < 0.01), indicating strengthened structural integrity. Chlorophyll content—a marker of photosynthetic efficiency—was significantly higher (*p* < 0.01). These findings collectively indicate that the consortia significantly influence multiple traits associated with plant growth and physiological performance.

### 2.3. Microbial Shifts After SynCom Treatment

Prior to SynCom treatment, the potting soil had the following physicochemical properties: pH 6.7, 4.2% organic matter content, and 15.4 meq/100 g of cation exchange capacity (CEC). Although no microbial profiling of the growth medium was performed before inoculation, the soil was collected from a previously used greenhouse bed, assumed to contain a moderately diverse native microbiota typical of greenhouse environments. The application of five-member SynCom significantly influenced microbial community composition across treatment groups and time points (Figure 2). At the phylum level (Figure 2a), Proteobacteria increased in the SynCom treatment group at day 7 (26.8% vs. 21.1%) but stabilized by day 60 (19.3% vs. 19.5%). Firmicutes were consistently enriched in the SynCom treatment group, while Acidobacteriota and Actinobacteriota were more abundant in the control group, particularly at day 60. Fungal communities were overwhelmingly dominated by Ascomycota (>95%) across SynCom group and time points. A slight decline to ~91% in the SynCom treatment group at day 7 indicated initial disturbance, followed by stabilization by day 60, suggesting a strong buffering capacity in the fungal community. At the class level (Figure 2b), Clostridia were significantly enriched (*p* < 0.05) in the SynCom treated samples, while Vicinamibacteria, Gemmatimonadetes, Planctomycetes, and Ktedonobacteria declined. Among fungi, Sordariomycetes and Eurotiomycetes were significantly more abundant in the SynCom treatment group (*p* < 0.05). Importantly, several bacterial genera corresponding to the applied strains—*Bacillus* spp. (strain T3), *Pseudomonas* spp. (strain A6), *Flavobacterium* spp. (strain T4), *Cytobacillus* spp. (strain T8), and *Streptomyces* spp. (strain T14)—were enriched in the SynCom treated rhizosphere, indicating successful establishment and viability of the introduced consortium. Additionally, fungal genera such as *Fusicolla* and *Schizothecium*, known for growth-promotion and pathogen suppression, were more abundant in SynCom treated samples. Although bulk soil microbial communities were analysed, the persistence of applied genera suggests effective colonization of the rhizosphere niche.

The diversity of microbial communities was assessed via alpha and beta diversity analyses to evaluate the effect of SynCom treatments on microbial structure. Alpha diversity indices, including Shannon (Figure 2d) and Chao1 (Figure 2e), revealed slight statistically significant differences in microbial richness and evenness between treated and control groups at day 7 and 60 post-treatment, suggesting comparable diversity across treatments. Conversely, the principal coordinates analysis (PCoA) based on Bray–Curtis dissimilarity showed significant shifts in microbial community composition between the SynCom treated and control groups (Figure 2f). Statistical validation via PERMANOVA further supported these findings, with R^2^ = 0.36 (*p* < 0.01) and R^2^ = 0.40 (*p* < 0.05) for bacterial and fungal communities, respectively. Collectively, these findings indicate that while the microbial richness and evenness remained unaffected, the SynCom treatment induced substantial changes in the overall microbial community composition.

### 2.4. Relationships Between Microbial Community and Soil Physicochemical Property

Environmental factors—including NO_2_^−^-N, NO_3_^−^-N, NH_4_^+^-N, K^+^, Ca^2+^, and pH—were analysed to explore their relationship with microbial communities under control and SynCom treatment conditions over 7 and 60 days (Figure 3a). A two-way ANOVA revealed a significant treatment-time interaction for NO_3_^−^-N (*p* < 0.001), with higher levels observed in the SynCom treated group, especially at day 60. Potassium (K^+^) showed a significant treatment effect (*p* < 0.01), with consistently higher concentrations in the SynCom treated group. Calcium (Ca^2+^) also showed a significant treatment effect (*p* < 0.05), with elevated levels in the SynCom treated group at both time points. Although NH_4_^+^-N and NO_2_^−^-N levels were higher in the SynCom treated group, these differences were not statistically significant (*p* > 0.05). Additionally, pH remained stable across treatments and time points, with no significant differences (*p* > 0.05). These shifts in nutrient concentrations are probably driven by SynCom treatments, which actively modulate nutrient cycling processes. Euclidean distance analysis demonstrated significant correlations between Bray–Curtis dissimilarity and the Euclidean distances of several environmental factors (Appendix A), highlighting the effect of altered microbiome owing to SynCom treatment. For bacterial communities, NH_4_^+^-N (R^2^ = 0.524, *p* < 0.0001) and NO_2_-N (R^2^ = 0.49, *p* < 0.0001) exhibited strong correlations, suggesting that shifts in ammonium and nitrite concentrations are influenced by microbiome-mediated changes. Similarly, in fungal communities, significant correlations were observed between Bray–Curtis dissimilarity and the Euclidean distances of several environmental factors, including Ca^2+^ (R^2^ = 0.261, *p* = 0.034), NH_4_^+^ -N (R^2^ = 0.37, *p* = 0.0022), NO_2_^−^-N (R^2^ = 0.267, *p* = 0.0302), and NO_3_^−^-N (R^2^ = 0.299, *p* = 0.0148). These findings indicate that the compositional changes within fungal communities, driven by SynCom-induced microbiome alterations, influenced soil chemistry dynamics.

Pearson correlation analysis was conducted to evaluate the strength of the relationships between environmental factors and microbial genera in bacterial and fungal communities (Figure 3b). In bacterial communities (top), key genera exhibited unique associations with environmental factors. *Bacillus* displayed significant positive correlations with NO_3_^−^-N, highlighting its role in nitrate-enriched environments under SynCom treatment. *Sphingomonas* were positively associated with NH_4_^+^-N and NO_2_^−^-N, indicating their involvement in nitrogen cycling, influenced by the altered microbiome. *Pseudolabrys* showed a positive association with pH, indicating its preference for more stable, untreated conditions. In fungal communities (bottom), similar patterns were observed, with *Mycosphaerella* and *Cladosporium* positively correlated with NO_3_^−^-N and Ca^2+^, suggesting their roles in environments with higher nitrate and calcium concentrations under the SynCom treatment. *Phialemonium* and *Mycosphaerella* also showed strong positive associations with K+ and NO_2_^−^-N, indicating that the SynCom treatment influenced the availability of K^+^ in the soil. Conversely, *Verticillium* and *Talaromyces* exhibited negative correlations with NH_4_^+^-N and NO_2_^−^-N, suggesting sensitivity to elevated nutrient levels introduced by the SynCom treatment.

Redundancy analysis (RDA) revealed the significant influence of environmental factors on the microbial community composition at the genus level for bacterial and fungal communities (Figure 3c). In bacterial communities (left), RDA showed that changes in environmental variables—such as NH_4_^+^-N, NO_2_^−^-N, NO_3_^−^-N, K^+^, Ca^2+^, and pH—were driven by the microbiome restructuring induced by the SynCom treatment, accounting for 39.1% and 27.3% of the variation on RDA1 and RDA2, respectively. The *Bacillus* was positively associated with elevated NO_3_^−^-N levels, highlighting its contribution to nitrate enrichment in the SynCom treated soils. Conversely, *Anaeromyxobacter* and *Citrifermentans* were associated with pH stability in control samples, highlighting their preference for untreated soil conditions. *Sporacetigenium* was associated with elevated K^+^, Ca^2+^, NH_4_^+^-N and NO_2_^−^-N levels, suggesting their roles in nitrogen cycling under treatment conditions. In fungal communities (right), RDA revealed that shifts in environmental factors—such as NO_3_^−^-N, NH_4_^+^-N, K^+^, Ca^2+^, and pH—were also strongly associated with fungal community composition, with RDA1 and RDA2 accounting for 33.2% and 29.7% of the variation, respectively. *Mycosphaerella* was associated with increased NO_3_^−^-N and Ca^2+^ concentrations in treated soils. This indicates that the altered fungal microbiome may contributed to nutrient change in the soil environment. The fungal genera, such as *Acremonium* and *Cladosporium,* showed the significant associations with NH_4_^+^-N and K^+^, emphasizing further the influence of fungal activity on nutrient redistribution under SynCom treatment.

### 2.5. Microbial Metabolic Changes After Syncom Treatment

To evaluate the ecological roles of bacterial and fungal communities under SynCom treatment, functional predictions and trait distribution analyses were conducted (Figure 4). The predicted functional pathways of bacterial communities (Figure 4a) revealed significant shifts in metabolic processes influenced by the altered microbiome owing to SynCom treatment. The SynCom Treated samples showed enrichment in key metabolic pathways, including nitrogen metabolism and the biosynthesis of secondary metabolites, suggesting enhanced nutrient cycling and microbial-driven resource utilisation. However, pathways related to glycosaminoglycan degradation and polyketide sugar unit biosynthesis were reduced in the SynCom treated samples. This indicates a shift away from these specific metabolic processes in response to the SynCom treatment. Conversely, pathways associated with cellular motility and biofilm formation were significantly increased, suggesting enhanced microbial activity and potential establishment of microbial networks in the SynCom treated soils. Fungal functional guilds (Figure 4b) showed significant shifts in relative abundances under the SynCom treatment. The abundance of soil saprotrophs increased in treated samples on day 7 and particularly on day 60, indicating enhanced decomposition activity and nutrient mobilization. Conversely, the relative abundance of plant pathogens decreased in treated soils, suggesting potential disease suppression effects induced by the altered fungal microbiome. No significant differences were observed in guilds—such as mycoparasites and foliar endophytes—indicating stability in these functional groups across the groups.

## 3. Discussion

### 3.1. Effect of SynCom Composed of 5 PGPR on Pepper Plant Resistance to Phytophthora capsici

Phytophthora blight, caused by *P. capsici*, poses a major threat to pepper cultivation, often resulting in significant yield losses. In this study, *P. capsici* infection markedly impaired pepper growth and productivity. However, treatment with a formulated PGPR-based SynCom significantly improved pepper plant resilience under *P. capsici* challenge, as evidenced by enhanced growth parameters and suppressed disease symptoms (Table 1; Figure 1c). Compared to the pathogen-only control, SynCom-treated plants exhibited substantial increases in height, root length, biomass, stem diameter, and chlorophyll content, all of which indicate vigorous growth and improved physiological status. This is consistent with previous studies demonstrating that complex root-associated microbial communities confer synergistic benefits to plant growth and stress resilience [2,6,15,16].

The SynCom members were specifically selected to maximize functional complementarity across key plant growth-promoting (PGP) activities, including phosphate solubilization, siderophore and IAA production, ammonia release, and multiple lytic enzyme activities (Table 1). The contribution of overlapping yet distinct PGP mechanisms is well established as a driver of improved plant performance under biotic stress. Notably, *Streptomyces roseicoloratus* T14 provided the highest direct antagonism to *P. capsici*, while T3, T4, A6, and T8 contributed to nutrient cycling, hormone modulation, and biofilm/cellulose activity, supporting both direct and indirect pathogen suppression. Consistent with these complementary roles, visual assessment and physiological measurements indicated that SynCom-treated plants retained higher chlorophyll levels and showed markedly less wilting and necrosis compared to untreated controls (Figure 1c). These protective effects parallel reports by [12,13], who showed that multi-strain inoculants having PGP-activities reduce disease severity and promote physiological vigor more effectively than single-strain treatments [3,6].

Microbiome profiling revealed successful rhizosphere colonization by the SynCom strains, as indicated by the increased relative abundance of *Bacillus*, *Streptomyces*, *Pseudomonas*, *Flavobacterium*, and *Cytobacillus* in treated soils (Figure 2c). Moreover, SynCom inoculation led to pronounced shifts in bacterial and fungal community composition (Figure 2f), with enrichment of beneficial genera such as *Fusicolla* and *Schizothecium*, and significant increases in soil nitrate and potassium concentrations (Figure 3a). These findings are in agreement with the ecological models proposed by [15,17], in which functionally diverse consortia drive community restructuring, improved nutrient cycling, and enhanced plant protection. Functional pathway analysis further demonstrated that SynCom treatment enriched pathways related to nitrogen metabolism, secondary metabolite biosynthesis, and biofilm formation, while reducing the relative abundance of predicted plant pathogenic fungal guilds (Figure 4a,b). This functional reprogramming of the rhizosphere microbiome aligns with mechanisms described in [3,17], who reported similar enhancements in nutrient mobilization and disease suppression following SynCom application [15,17].

Taken together, our results demonstrate that the ecological cooperation among SynCom members, underpinned by complementary PGP traits and stable rhizosphere colonization, confers multi-dimensional and resilient plant protection against *P. capsici*. These findings substantiate and extend recent literature emphasizing the superiority of microbial consortia over single-strain inoculants for sustainable disease management [2,15,16,18,19].

### 3.2. Shifts in Rhizosphere Microbial Diversity Under Pathogen Pressure

Bacterial communities in the rhizosphere are generally observed to exhibit heightened sensitivity to biotic stresses, such as pathogen invasion, leading to more pronounced shifts in their composition. Conversely, fungal community structures tend to remain relatively stable under similar conditions [20]. Consistent with these observations, our study revealed that the SynCom application induced significant changes in rhizosphere microbial community composition, as evidenced by beta diversity analysis. PCoA revealed distinct clustering between treated and control groups, indicating that the SynCom actively modulated the rhizosphere microbiome. These findings are consistent with a previous study demonstrating that PGPR inoculation alters microbial community structures, often enriching beneficial taxa while suppressing potential pathogens [21]. Conversely, alpha diversity indices such as Shannon and Chao1 showed slight significant differences in microbial richness or evenness, suggesting that the observed shifts were functional rather than compositional in nature. The enrichment of beneficial bacterial taxa, such as *Bacillus*, as well as fungal genera, including *Fusicolla* and *Schizothecium*, further highlights the positive effect of the SynCom on rhizosphere ecology. These genera are known for their roles in nutrient cycling, pathogen suppression, and plant health promotion. Moreover, the stabilization of microbial diversity over time suggests that the SynCom not only triggered immediate changes but also facilitated long-term ecological balance, a critical factor for sustaining plant health under pathogen pressure [22]. Above the direct action of the SynCom members, it is plausible that their introduction altered the rhizosphere microbiota in a way that enhanced systemic plant resistance. Notably, beneficial taxa such as *Bacillus* and *Trichoderma*—enriched in the SynCom-treated soils—are well known for their antagonistic activity against soil-borne pathogens. These taxa, although not originally introduced as part of the SynCom in the case of *Trichoderma*, may have been stimulated by shifts in the microbial niche and resource availability. Such indirect effects underscore the importance of microbiome restructuring in disease mitigation. Previous studies have shown that native bacterial consortia can synergize with arbuscular mycorrhizal fungi to enhance nutrient availability and plant immunity even under nutrient-poor soil conditions, further emphasizing the ecological role of rhizosphere composition in disease tolerance [5]. While individual SynCom strains were not quantified using strain-specific markers, several genera corresponding to the applied strains—including *Bacillus*, *Pseudomonas*, *Flavobacterium*, *Streptomyces*, and *Cytobacillus*—were enriched in the rhizosphere microbiome of SynCom-treated plants at both 7 and 60 DPI (Figure 2c). This suggests successful colonization and persistence of the SynCom members over time. The sustained presence of these genera supports the notion of functional stability and ecological integration of the SynCom within the existing microbial community, contributing to its observed effects on plant growth and disease suppression.

### 3.3. Functional Roles of Key Microbial Genera in Disease Resistance

The differential composition of dominant rhizosphere microbial communities at various time points appears to be a key factor influencing the responses of plants to *P. capsici* infection. Functional analysis of the rhizosphere microbiome demonstrates the essential roles of specific microbial genera in conferring disease resistance. The enrichment of *Paenibacillus*, known for its robust production of antimicrobial lipopeptides [23], probably contributed to the observed suppression of *P. capsici*. Similarly, the increase in *Ramlibacter*, associated with the production of secondary metabolites such as pyocyanin and phenazines, would enhance the antifungal efficacy of the SynCom treatment. The presence of *S. roseicoloratus* T14 within the SynCom, characterized by strong chitinase activity, may contribute to pathogen suppression by degrading chitin-containing structures of co-occurring fungal pathogens or by triggering plant defense responses via chitin fragments, despite *P. capsici* lacking chitin in its cell wall.

### 3.4. Metabolic Activities and Transport Mechanisms Supporting Disease Resistance

The predicted functional pathways derived from microbial community analysis revealed a pronounced enrichment in nitrogen metabolism and biosynthesis of secondary metabolites within the SynCom treated group. These findings closely align with the dynamic changes in soil nutrient profiles. Nitrogen cycling processes, predominantly driven by bacterial genera such as *Bacillus*, were strongly correlated with elevated concentrations of nitrate and ammonium in the rhizosphere. These changes indicate enhanced nutrient availability, which provides essential resources for sustaining plant growth, even under biotic stress conditions induced by pathogens [24]. The enrichment of these nitrogen-related pathways suggests a robust microbial adaptation to the altered rhizosphere environment, actively contributing to maintaining nutrient homeostasis and plant vitality [25]. Beyond nutrient-related processes, microbial pathways associated with biofilm formation and cellular motility were significantly enriched in the SynCom treated soils, suggesting enhanced microbial establishment and colonization within the rhizosphere [26]. Biofilm-forming species—including *Pseudomonas*, *Bacillus* and *Streptomyces*—are well-documented for their ability to create structured microenvironments that promote microbial persistence and functional efficiency. These biofilms can act as physical barriers, protecting plant roots from direct pathogen invasion while also facilitating the exchange of critical metabolites between the microbial community and the host plant [27]. Conversely, cellular motility is essential for effective root colonisation and the strategic positioning of beneficial microbes in proximity to plant tissues. It enables optimal interaction and mutualistic benefits [28]. The interplay between microbial functional adaptations and rhizosphere nutrient dynamics observed in the SynCom treated group highlights a complex network of synergistic interactions. These interactions collectively enhance the capacity of the rhizosphere to withstand pathogenic pressures and support plant health under stress conditions. The observed shifts in microbial metabolic activities suggest a concerted effort by the rhizosphere community to optimize resource acquisition and promote plant resilience. The SynCom included strains capable of solubilizing phosphate (e.g., *B*. sp T3, *S. roseicoloratus* T14), producing ammonia and urease (e.g., *S. roseicoloratus* T14, *B.* sp. T3), and synthesizing siderophores for iron acquisition (e.g., T14, T4). These metabolic activities enhanced rhizosphere nutrient status and likely contributed to the observed improvement in plant growth and chlorophyll content. In particular, SynCom-induced increases in nitrate and ammonium levels, as illustrated in Figure 3a, confirm microbial modulation of nitrogen cycling. These findings highlight the potential for targeted microbiome management to mitigate the effects of soil-borne pathogens in agricultural systems [29].

### 3.5. Study Limitations and Future Directions

While the current study provides important insights, certain limitations should be acknowledged. First, results were derived from a single experiment; however, each pot contained two pepper seedlings harvested at different timepoints (7 and 60 DPI), and the consistency observed across seven biological replicates strengthens the validity of the findings. Second, the absence of a SynCom-only control limits the ability to definitively attribute observed effects to disease suppression rather than plant growth promotion alone. Third, direct quantification of disease severity, such as lesion scoring or pathogen load measurement, was not performed in this study. Instead, physiological indicators—including plant biomass, chlorophyll content, and visual symptoms—were used as indirect proxies. The observed reduction in wilting and lesions in SynCom-treated plants Appendix A supports the conclusion that the SynCom contributed to pathogen mitigation. Future studies should incorporate SynCom-only treatments alongside direct disease assessments to more clearly disentangle these effects. Although fruit yield was not assessed in this study due to the pepper cultivar’s growth cycle (fruiting begins after ~80 days), early-stage improvements in biomass, chlorophyll content, and root-shoot development are well-established predictors of higher reproductive success and yield in pepper [30]. The PGPR traits present in our SynCom—such as nitrogen cycling, IAA production, and phosphate solubilization—have been linked to increased fruit yield in multiple crop systems [31]. Therefore, the physiological enhancements observed here likely translate into improved yield potential under pathogen stress.

## 4. Materials and Methods

### 4.1. Isolation and Characterization of Plant Growth-Promoting Rhizobacteria

The bacterial strains were isolated from the rhizosphere soil of pepper plants on 24 January 2021 from greenhouse facilities at Kyungpook National University, Daegu, Republic of Korea (35°53′40″ N, 128°36′49″ E). During sampling, the environmental conditions were as follows: soil temperature was 15 °C, moisture content was 18%, and soil pH was 6.5. Rhizosphere soil was obtained by carefully uprooting healthy pepper plants and gently shaking the roots to remove loosely attached bulk soil; approximately 1 g of soil that remained tightly adhered to the roots was collected using sterile forceps. This soil was then suspended in 100 mL of sterile distilled water for serial dilution up to a 6-fold concentration. Aliquots from these dilutions were spread onto tryptic soy agar (TSA) plates (Difco Laboratories, Detroit, MI, USA) and incubated at a constant temperature of 30 °C. After 48 h, colonies exhibiting diverse morphologies began to emerge [32].

Biochemical assays were conducted to determine the plant growth-promoting traits of the bacterial isolates as follows: IAA production was evaluated using Luria–Bertani (LB) broth supplemented with L-tryptophan, followed by colorimetric analysis with Salkowski reagent. Optical density was measured at 530 nm, and IAA concentrations were calculated using a standard curve [33]. Phosphate solubilisation was assessed on the National Botanical Research Institute’s phosphate (NBRIP) agar medium, using tricalcium phosphate as the phosphorus source. The halo zone diameters were measured, and the phosphate solubilisation index was calculated [34]. Siderophore production was analysed using the Chrome Azurol S assay, where the presence of orange halo diameters around colonies was recorded, and the siderophore production index was calculated [35]. Ammonia production was determined by incubating the isolates in peptone broth. After incubation, the supernatant was treated with Nessler’s reagent, and a yellow or brown change in colour indicated ammonia production, a marker of enhanced nitrogen availability for plants [36]. Catalase activity was confirmed by observing oxygen bubble release following exposure to hydrogen peroxide, indicating the ability to detoxify reactive oxygen species [37]. Chitinase activity—indicating the ability to degrade fungal cell wall components—was evaluated using colloidal chitin agar plates. Clear zones around colonies signified chitinase activity [38]. Antifungal activity was assessed via dual culture confrontation assays. A 5-mm plug of actively growing *P. capsici* mycelium was placed on V8 agar opposite the bacterial isolates. After 7 days of incubation at 30 °C, the inhibition zone was measured to determine the antagonistic potential of each isolate [32]. Biofilm formation was analysed using the Congo red agar method. Black, dry crystalline colonies indicated biofilm production, whereas red or pink colonies suggested non-biofilm-forming strains [39]. Cellulase activity was evaluated using carboxymethyl cellulose agar plates. Clear halos around colonies after flooding with Congo red solution followed by a NaCl rinse indicated cellulase activity. Hydrolysis capacity was quantified by measuring the halo and colony diameters [40]. These plant growth-promoting traits—such as siderophore production, IAA biosynthesis, and chitinase activity—were experimentally verified through in vitro assays for individual isolates prior to SynCom formulation. However, these traits were not directly re-quantified in the rhizosphere soil following plant inoculation.

### 4.2. SynCom Design

In forming the SynCom, five PGPR strains were selected based on beneficial plant growth-promoting traits and functional complementarity, although minor overlaps in functional activities between strains were observed. *Streptomyces roseicoloratus* T14, a positive strain for production of protease and chitinase was specifically chosen for its antifungal activity against *P. capsici*. The other strains were included for their abilities to produce IAA, solubilise phosphate, synthesize ammonia, and produce siderophores to facilitate iron uptake [41]. Strains were tested for compatibility using dual-culture assays on nutrient agar plates [42]. Each strain pair was co-cultured to check for inhibition zones or overgrowth, and only non-antagonistic combinations were included in the final consortium.

For molecular characterisation based on taxonomic identification, the bacterial 16S rRNA gene was targeted. Genomic DNA was extracted using the Wizard™ Genomic DNA Purification Kit (Promega Corporation, Madison, WI, USA) following the manufacturer’s protocol. The 16S rRNA gene was amplified via PCR using universal primers 27F (5′-AGAGTTTGATCATGGCTCAG-3′) and 1492R (5′-TACGGYTACCTTGTTACGACTT-3′), with sequencing conducted using primers 785F (5′-GGATTAGATACCCTGGTA-3′) and 907R (5′-CCGTCAATTCMTTTRAGTTT-3′) on an Applied Biosystems 3730XL system at Macrogen Inc. (Seoul, Republic of Korea). The sequences were cleaned of ambiguous nucleotides, analysed via BLAST + version 2.11.0 against GenBank (http://www.ncbi.nlm.nih.gov (accessed on 27 March 2021)). and deposited with accession numbers: *Bacillus* sp. T3 (PP763279), *F. anhuiense* T4 (PP763424), *C. firmus* T8 (OR857502), *S. roseicoloratus* T14 (PP763444), and *P. frederiksbergensis* A6 (PP763475).

The growth kinetics of these strains were monitored over a 12-h period. Cultures were standardized to an initial optical density (OD) of 0.1 at 600 nm in Luria-Bertani (LB) broth containing 1% NaCl, and incubated at 30 °C with shaking at 200 rpm [43]. OD_600_ readings were recorded hourly, and corresponding colony-forming units per millilitre (CFU/mL) were determined by serial dilution plating on LB agar. Growth curves were plotted based on CFU data, and specific growth rates were calculated for each strain [44].

### 4.3. Pathogenicity of Phytophthora capsici

*P. capsici* KACC 40,157 was obtained from the National Agrobiodiversity Center Genebank. Sporangial suspensions were prepared, and pathogenicity assays were conducted using dual-culture methods to evaluate the antagonistic activity of the PGPR strains [45]. The percentage inhibition of mycelial growth was calculated using the formula:$$Inhibition\;\%=\left(\frac{Control\;radial\;growth-treatment\;radial\;growth}{Control\;radial\;growth}\right)\times 100$$

### 4.4. Experimental Design and Soil Sample Collection

The experiment was conducted in a controlled growth chamber at Kyungpook National University under day/night temperatures of 18–23 °C and 12–16 °C, respectively, with consistent artificial light. Eighteen-day-old pepper seedlings (*Capsicum annuum* L.) were transplanted individually into plastic pots (1 plant/pot) containing 1.2 kg of sieved potting soil collected from the greenhouse where the bacterial strains were originally sampled. A completely randomized design (CRD) was used with seven biological replicates per treatment group, and the experiment was conducted once. This number of replicates was chosen based on power analysis (95% confidence, 80% power) and growth chamber space limitations. Treatment groups included: (1) pathogen (inoculated with *P. capsici*) and (2) SynCom (inoculated with *P. capsici* and the SynCom). The *P. capsici* isolate (KACC 40170) was cultured on V8 agar at 25 °C for 7 days. Zoospores were obtained by flooding the plates with sterile distilled water and adjusting the suspension to ~10^6^ zoospores/mL. Each plant received a 20 mL soil drench of this suspension at transplanting. The SynCom consisted of five strains cultured individually in TSB to a final density of ~10^6^ CFU/mL. After incubation, strains were harvested from their respective media by centrifugation and resuspended in sterile distilled water. Equal volumes of each resuspended culture were then combined, and the mixture (20 mL per plant) was applied as a soil drench at transplanting and again two weeks later to ensure rhizosphere colonization. This concentration was selected to reflect more achievable inoculum levels for practical biocontrol applications. No fertilizer was applied during the experiment to avoid altering microbiome structure. Soil samples were collected at two time points: day 7 post-inoculation (to observe early microbial shifts), and day 60 (to assess longer-term changes). At 60 days, pepper plants had reached the early flowering stage, a critical period for analysing microbiome-plant-pathogen interactions [46].

### 4.5. Evaluation of Plant Growth Property

The experiment was conducted in a controlled growth chamber at Kyungpook National University under day/night temperatures of 18–23 °C and 12–16 °C, respectively, with consistent artificial light. Eighteen-day-old pepper seedlings (*Capsicum annuum* L.) were transplanted individually into plastic pots (1 plant/pot) containing 1.2 kg of sieved potting soil collected from the greenhouse where the bacterial strains were originally sampled. Due to the cultivar’s biological growth cycle, with fruiting initiation starting approximately 80 days after transplanting, fruit yield could not be assessed. Pots contained two plants each: one was harvested at day 7 for early-stage analysis and the second at day 60 for final evaluation of vegetative growth parameters.

### 4.6. Bacterial and Fungal Amplicon Sequencing

Genomic DNA was extracted from rhizosphere soil samples collected at day 7 and day 60 post-inoculation using the DNeasy PowerSoil Kit (Qiagen, Hilden, Germany) following the manufacturer’s protocol [47]. DNA concentration was measured using a Qubit fluorometre (Thermo Fisher, Waltham, MA, USA). For library preparation for 16S rRNA sequencing, the V4 region of the 16S rRNA gene was amplified using a two-step PCR approach. The first PCR utilized 515F/806R primers, with an initial denaturation at 95 °C for 3 min, followed by 25 cycles of 95 °C for 30 s, 55 °C for 30 s, and 72 °C for 30 s. For ITS amplicon sequencing, the ITS1 region was amplified using ITS1F/ITS2 primers in a similar two-step PCR process, as stated earlier. A second PCR incorporated index primers with 8 cycles under similar conditions for 16S rRNA and ITS amplicon sequencing. Libraries were purified using AMPure XP beads (Beckman Coulter, Brea, CA, USA) and quality-checked on an Agilent 2100 Bioanalyzer (Agilent Technologies, Santa Clara, CA, USA). Sequencing was conducted on an Illumina MiSeq platform using a MiSeq Reagent Kit v2 (300 bp single-end reads) (Illumina, Inc., San Diego, CA, USA) at KNU NGS Core Facility (Daegu, Republic of Korea) [48]. While PCR-based quantification of *P. capsici* was not performed in this study, its impact was inferred through disease symptom observation and associated microbiome shifts.

### 4.7. Bioinformatics Analysis

Raw amplicon sequence data were analysed using the Qiime2 analysis pipeline (version 2024.02). The raw sequences were initially imported into the Qiime2 platform, where quality control procedures were implemented using the DADA2 algorithm. This step involved filtering, denoising, and removing chimeric sequences to ensure high quality and accurate reads. This process led to high-quality reads across all samples, which were then prepared for downstream analysis [49]. The DADA2 algorithm enabled the generation of amplicon sequence variants, offering more resolution compared to that of traditional OTU-based approaches. Following sequence filtering, taxonomy was assigned to bacterial sequences using the SILVA reference database (SILVA SSU version 138.1). For fungal sequences, taxonomy was determined using the UNITE database (UNITE ver10.99 classifier 04.2024).

### 4.8. Functional Prediction of Microbial Communities

Functional predictions for bacterial communities were conducted using Phylogenetic Investigation of Communities by Reconstruction of Unobserved States [50]. Kyoto Encyclopaedia of Genes and Genomes (KEGG) orthologues were employed for functional annotations, focusing on pathways related to metabolism, environmental information processing, and cellular processes. Functional changes were visualized via graphical representations, displaying significant pathway alterations in terms of relative abundance and fold changes. For fungal communities, functional traits were inferred using the FUNGuild database [51], which classifies taxa into trophic modes and ecological roles such as saprotrophic, symbiotic, and pathogenic fungal guilds. Differences in guild composition and relative abundance between control and treated groups were also evaluated.

### 4.9. Assessment of Soil Physicochemical Properties

Soil samples from each pot were collected at 10 cm depth using a sterile spatula, with three subsamples per pot pooled into one composite sample per treatment for analysis. To extract the soil pore water, 5 g of each composite sample was mixed with 25 mL of sterile distilled water (SDW), vortexed, and incubated at 30 °C for 1 h. Samples were then sonicated for 10 min using a Branson 8210 ultrasonic bath (Branson Ultrasonics Corp., Danbury, CT, USA) at 40 kHz to enhance solubilization of ions. The suspensions were filtered through 0.20 μm cellulose acetate syringe filters (Hyundai Micro, Seoul, Republic of Korea, SC25P020SS) using sterile 10 mL syringes (Kovax Syringe, Gangnam, Seoul, Republic of Korea). The filtrates were collected in corresponding vials and analyzed for ion concentrations (NO_2_^−^, NO_3_^−^, NH_4_^+^, K^+^, and Ca^2^^+^) using a Thermo Scientific Dionex ICS-5000 ion chromatography system equipped with an AS autosampler and DC detector module (Thermo Fisher Scientific, Gangnam, Seoul, Republic of Korea). Soil pH was measured from separate fresh subsamples using a pH meter after mixing soil with SDW at a 1:2.5 (*w*/*v*) ratio and allowing the suspension to equilibrate for 30 min prior to measurement [52].

### 4.10. Statistical Analysis

All statistical analyses were conducted using R software (version 4.4.1; http://www.r-project.org/ (accessed on 30 August 2024)). Taxonomy files generated from Qiime2 were imported into R for statistical evaluation and visualisation using the microeco package (version 1.8.0). Alpha diversity metrics, including Shannon and Chao1 indices, were calculated to assess microbial richness and diversity via two-way ANOVA. Beta diversity was evaluated based on Bray–Curtis dissimilarity, with ordination conducted using PCoA. The statistical significance of beta diversity differences was tested using PERMANOVA (vegan package, version 2.6–6.1). The relative abundance and taxonomic composition at the phylum and class levels were analysed using the phyloseq package (version 1.48.0). A two-way ANOVA was conducted to determine statistical significance in class-level composition, followed by Tukey’s HSD test for post-hoc analysis where applicable. Correlation analysis between microbial taxa and experimental variables was conducted using Spearman’s correlation (Hmisc package, version 5.1-3). RDA was employed to examine the relationships between microbial community composition and environmental factors (vegan package, version 2.6–6.1). Pearson correlation between environmental variables and microbial genera was also calculated using the Hmisc package. All statistical analyses were considered significant at *p* < 0.05.

## 5. Conclusions

This study offers compelling evidence for the dual role of SynCom using PGPR in promoting plant growth while suppressing soil-borne pathogens, particularly *P. capsici*. By leveraging the complementary activities of five bacterial strains, the SynCom not only reduced disease severity but also enhanced plant growth parameters and reshaped the rhizosphere microbiome. The enrichment of beneficial microbial taxa and functional pathway modulation demonstrates the potential of such SynCom to offer a sustainable alternative to chemical inputs in agriculture. However, limitations to this study should be considered. The experiment was conducted under controlled conditions with managed humidity to minimize pathogen spread, which may not fully replicate the complexities of field environments. The results may, therefore, be influenced by factors such as temperature, soil composition, and microbial interactions, which vary in natural settings. Future field trials under diverse environmental conditions are essential to validate these findings and to further investigate the ecological and economic benefits of SynCom in integrated disease management strategies.

## Figures and Tables

**Figure 1 plants-14-01625-f001:**
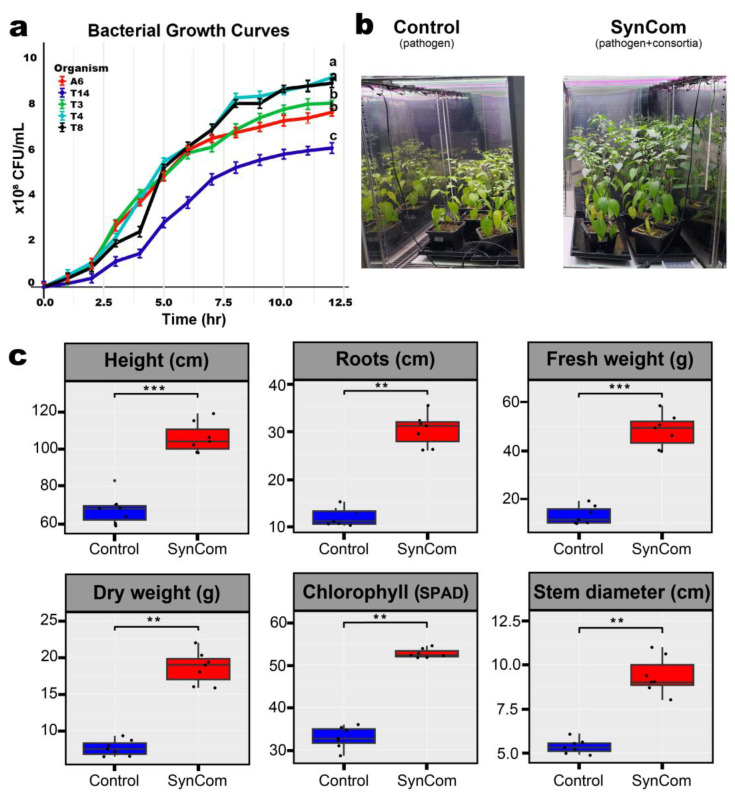
Bacterial growth and pepper plant performance under different treatments (Pathogen: *Phytophthora capsici*-treated, SynCom: *P. capsici* + 5-member SynCom) (**a**) Growth curves of *Pseudomonas frederiksbergensis* A6, *Streptomyces roseicoloratus* T14, *Bacillus* sp. T3, *Flavobacterium anhuiense* T4, and *Cytobacillus firmus* T8 over 12 h under standard laboratory conditions. Data are represented as means ± SE (n = 3). Different letters at 12-h mark indicate significant differences (*p* < 0.05 ANOVA, Tukey’s test). (**b**) Pepper plants at 60 days post-inoculation under pathogen and pathogen + SynCom conditions. (**c**) Boxplots of plant growth parameters: height, root length, fresh/dry weight, chlorophyll (SPAD), and stem diameter. Data represent means ± SD (n = 7). Asterisks indicate significance (** *p* < 0.01, *** *p* < 0.001; two-tailed Student’s *t*-test).

**Figure 2 plants-14-01625-f002:**
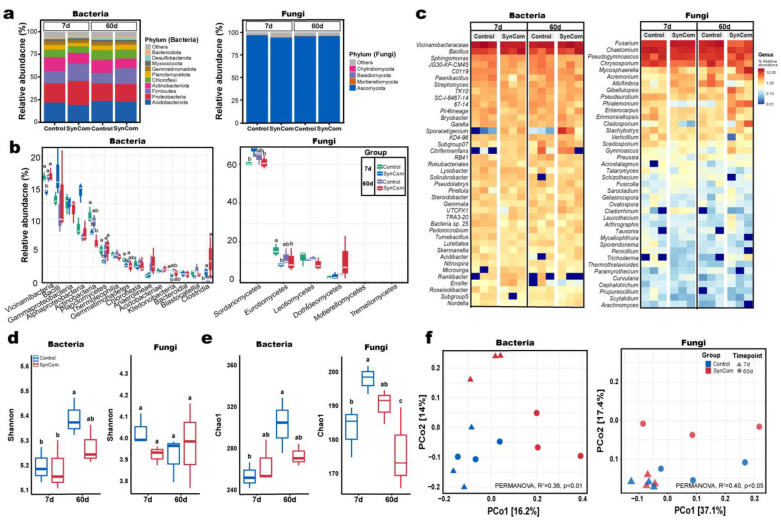
Microbial community composition and diversity in the rhizosphere of pepper plants under pathogen treatment and pathogen + SynCom treatment conditions at 7- and 60-days post inoculation. (**a**) Stacked bar plots of dominant bacterial and fungal phyla. (**b**) Boxplots showing class-level abundance differences; different letters indicate significance (*p* < 0.05; ANOVA, Tukey’s test). (**c**) Genus-level heatmaps of bacterial (**left**) and fungal (**right**) communities. (**d**) Shannon and (**e**) Chao1 diversity indices showing community richness and evenness; different letters indicate significance (*p* < 0.05; two-way ANOVA, Tukey’s test). (**f**) Principal coordinate analysis based on Bray–Curtis dissimilarity showing community shifts over time and treatments; PERMANOVA R^2^ and *p*-values are shown.

**Figure 3 plants-14-01625-f003:**
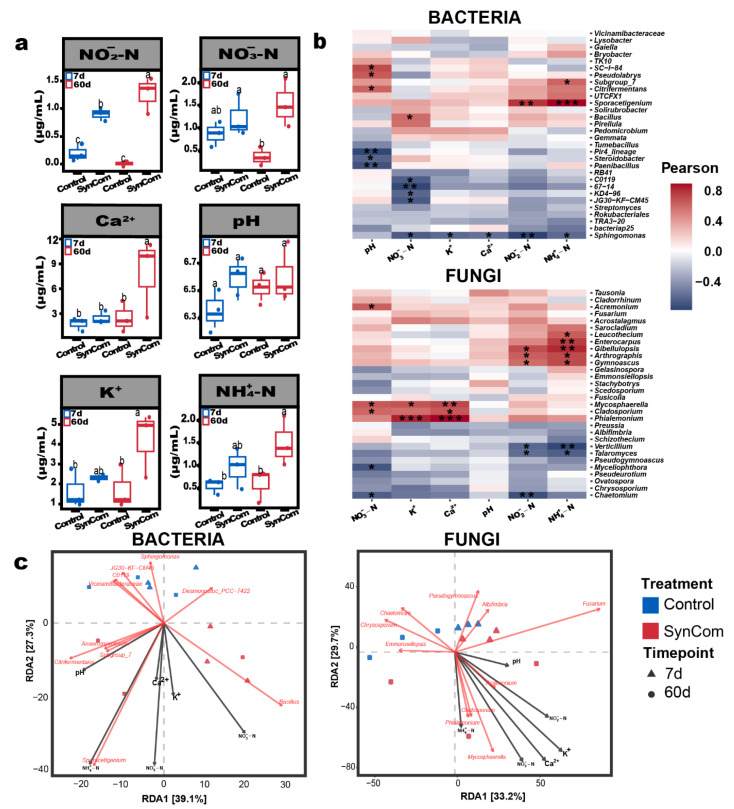
Correlation of environmental parameters with bacterial and fungal communities in the rhizosphere pathogen and pathogen + SynCom conditions over time (7 and 60 days). (**a**) Boxplots showing significant differences in NO_2_^−^-N, NO_3_^−^-N, K^+^, Ca^2^^+^, pH, and NH_4_^+^-N. Different lowercase letters above boxplots indicate statistically significant differences between groups at *p* < 0.05, based on Tukey’s HSD test. Groups sharing at least one letter (e.g., “ab”) are not significantly different, while groups with different letters (e.g., “a” vs. “b” or “a” vs. “c”) are significantly different. (**b**) Heatmaps of Pearson correlations between environmental parameters and microbial genera, with significance (* *p* < 0.05, ** *p* < 0.01, *** *p* < 0.001). (**c**) Redundancy analysis plot illustrating environmental influences on bacterial and fungal communities.

**Figure 4 plants-14-01625-f004:**
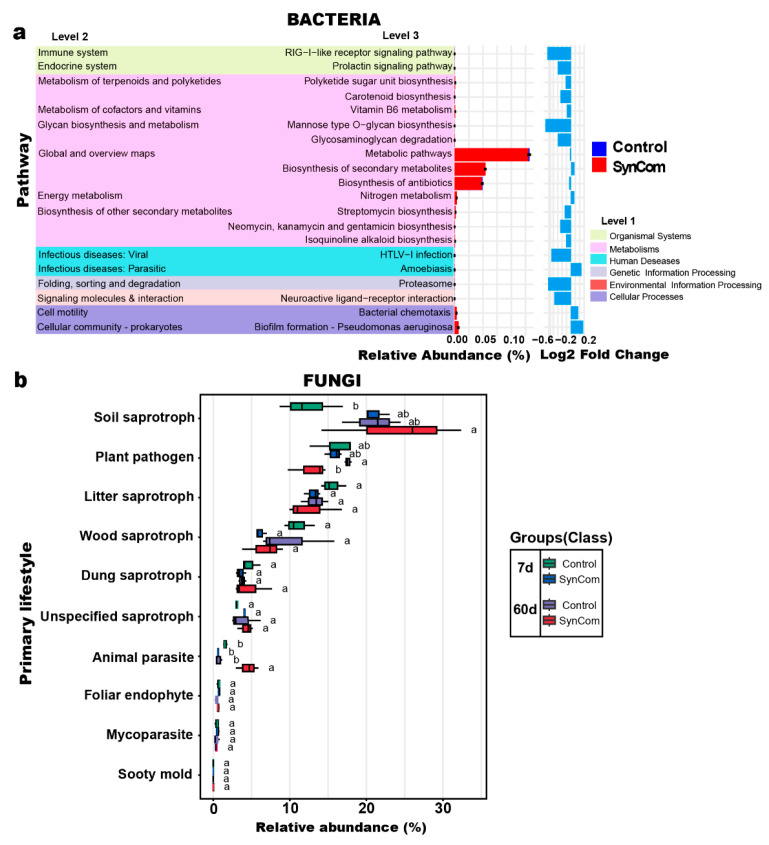
Functional analysis of bacterial and fungal communities in the rhizosphere of pepper plants under pathogen and pathogen + SynCom conditions. (**a**) Functional pathway analysis for bacterial communities based on KEGG Level 1, 2, and 3 categories. Bar plots showing relative abundance (%) of predicted pathways and Log2 fold changes between treatments. (**b**) Primary lifestyles of fungal communities, including soil saprotroph, plant pathogen, litter saprotroph, wood saprotroph, dung saprotroph, and others. Boxplots showing relative abundance (%) across treatments and time points (7 and 60 days), with letters indicating statistical differences (*p* < 0.05, ANOVA followed by Tukey’s post-hoc test).

**Table 1 plants-14-01625-t001:** Quantitative measurement of antifungal activity, enzymatic activities, and plant growth-promoting activities of the SynCom-five members. Data are presented as means ± standard deviations (n = 3). Different superscript letters within a row indicate statistically significant differences among bacterial strains for the corresponding trait (*p* < 0.05, Tukey’s HSD test).

Plant Growth Promoting Activity	*Bacillus* sp. T3	*F. anhuiense* T4	*C. firmus* T8	*S. roseicoloratus* T14	*P. frederiksbergensis* A6
Antifungal *P. capsici* (% inhibition)	0.03 ± 0.01 ^c^	0.001 ± 0.001 ^c^	0.007 ± 0.002 ^c^	58.4 ± 3.2 ^a^	17.6 ± 2.5 ^b^
Chitinase (mm halo zone)	3.5 ± 0.5 ^b^	0.005 ± 0.002 ^c^	0.06 ± 0.001 ^c^	12.3 ± 1.8 ^a^	0.002 ± 0.001 ^c^
Siderophore (mm reaction zone)	0.002 ± 0.001 ^c^	7.2 ± 0.6 ^b^	0.002 ± 0.001 ^c^	11.5 ± 0.8 ^a^	0.003 ± 0.0001 ^c^
Protease (mm halo)	2.1 ± 0.4 ^b^	0.003 ± 0.001 ^c^	3.4 ± 0.5 ^b^	9.7 ± 0.9 ^a^	0.0001 ± 0.0005 ^c^
IAA (μg/mL)	10.5 ± 1.2 ^b^	18.7 ± 1.5 ^a^	7.4 ± 0.8 ^c^	12.8 ± 1.1 ^ab^	8.3 ± 0.9 ^c^
Phosphate solubilization (PSI, NBRIP)	4.4 ± 0.7 ^a^	0.0014 ± 0.0005 ^c^	0.0012 ± 0.0003 ^c^	3.8 ± 0.5 ^ab^	2.7 ± 0.6 ^b^
Ammonia production (μg/mL)	2.0 ± 0.3 ^b^	2.5 ± 0.4 ^b^	2.5 ± 0.4 ^b^	4.8 ± 0.7 ^a^	2.0 ± 0.3 ^b^
Starch (mm reaction zone)	0.0009 ± 0.0002 ^c^	4.7 ± 0.8 ^ab^	5.8 ± 0.9 ^a^	0.002 ± 0.0003 ^c^	3.2 ± 0.6 ^b^
Cellulose activity (mm halo)	7.3 ± 0.6 ^a^	6.9 ± 0.5 ^a^	6.7 ± 0.6 ^a^	7.2 ± 0.4 ^a^	7.5 ± 0.5 ^a^
Glucose utilization (OD_600_)	0.78 ± 0.2 ^a^	0.91 ± 0.2 ^a^	0.52 ± 0.1 ^a^	0.93 ± 0.2 ^a^	0.95 ± 0.2 ^a^
Urease (mm reaction zone)	45.6 ± 3.4 ^a^	0.0 ± 0.0 ^d^	15.7 ± 1.5 ^c^	20.4 ± 2.2 ^b^	12.3 ± 1.4 ^c^

## Data Availability

The data that support the findings of this study are openly available in the NCBI Sequence Read Archive (SRA) and NCBI BioProject under BioProject number PRJNA1203901 at https://www.ncbi.nlm.nih.gov/bioproject/PRJNA1203901 (accessed on 30 March 2025) and https://www.ncbi.nlm.nih.gov/sra (accessed on 30 March 2025), with amplicon sequencing data accession numbers SRR31850677-SRR31850688 for 16S rRNA and SRR31850842-SRR31850853 for ITS. Bacterial strains used in the study include *Bacillus* sp. T3 (GenBank accession: PP763279), *Flavobacterium anhuiense* T4 (GenBank accession: PP763424), *Cytobacillus firmus* T8 (GenBank accession: OR857502), *Streptomyces roseicoloratus* T14 (GenBank accession: PP763444), and *Pseudomonas frederiksbergensis* strain A6 (GenBank accession: PP763475) at https://www.ncbi.nlm.nih.gov/nuccore (accessed on 30 March 2025). The pathogen *Phytophthora capsici* strain KACC 40147 is available in the Korea GenBank at https://genebank.rda.go.kr/microbeSearchView.do?sFlag=ONE&sStrainsn=6035 (accessed on 30 March 2025).

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
