# Peer review of "Application of a Synthetic Microbial Community to Enhance Pepper Resistance Against Phytophthora capsici"

_plants, 2025, doi:10.3390/plants14111625_

Round 1
Reviewer 1 Report
Comments and Suggestions for Authors
The manuscript by Bashizi and co-workers reports the results of an experiment in which a synthetic microbial community (5 members from 4 different bacterial classes) isolated from the rhizosphere of pepper plants was used to determine its effectivenesses in suppressing disease caused by Phytophthora capsici.
Major concerns
- All results from single experiment, seven biological replicates.
- The plant experiments lack a SynCom-only inoculate control: apparent ability of SynCom to suppress disease might be due totally to its plant growth-promoting effects.
- SynCom composition was not based on unique abilities shown by the five members, as stated. Strain T14 is superior or equal to others in a number of traits, e.g., it's equal to T3 in phosphate solubilization, higher than T4 in siderphore production, higher than A6 in IAA production and higher than T8 in ammonia production. This negates the stated rational for including these other strains based on them supposedly having "unique" traits. SynCom was not "specifically formulated" (line 301 and Section 2.1), in my opinion. This is a serious problem and pretty much negates making conclusions about disease suppression or changes in community structure/predicted metabolic changes.
- Actual ability of SynCom to reduce disease severity was not directly addressed, only plant mass, length and chlorophyll contents were determined. From the photo in Fig. 1, the pathogen-only plants don't look severely wilted, but the photos are worthless for seeing any other differences.
- Much of the text in figures (e.g., Fig. 2 b, c, f) is illegible, blow-up of pdf destroys resolution so it's not an option for reading the text.
Author Response
Reviewer #1
General comments: The manuscript by Bashizi and co-workers reports the results of an experiment in which a synthetic microbial community (5 members from 4 different bacterial classes) isolated from the rhizosphere of pepper plants was used to determine its effectiveness in suppressing disease caused by Phytophthora capsici.
Response: We thank the reviewer for the thoughtful and critical feedback. We have carefully revised the manuscript to address each of the concerns raised. Notably, we clarified the rationale for SynCom construction (Section 4.2), acknowledged the absence of a SynCom-only control as a limitation (Section 4.6), and expanded our discussion of the SynCom’s individual strain traits and potential functional redundancies. We also clarified the inability to measure yield due to the plant's biological cycle (Section 2.3) and acknowledged that the plant health effects were assessed via growth and physiological proxies, not direct lesion scoring (Section 4.6). Additionally, the figure resolution has been enhanced. Please find our detailed responses below.
Specific comments
Comment 1: All results from a single experiment, seven biological replicates.
Response: We thank the reviewer for raising this important point. While the study was based on a single experimental run, each treatment included seven biological replicates, and each pot contained two pepper seedlings. Importantly, we designed the experiment to capture temporal variation: one seedling per pot was collected at 7 days post-inoculation (DPI) for early-stage analysis, and the second seedling was collected at 60 DPI to assess growth and disease impact closer to the flowering stage. This staggered design enabled us to evaluate both short-term microbial effects and longer-term plant responses, thereby enhancing the interpretive value of the dataset. We have clarified this aspect in the revised manuscript and acknowledged the limitation of single-run experimentation (Line 393-398).
Revised text (Section 3.5, Line 433-438):
“While the current study provides important insights, certain limitations should be acknowledged. First, results were derived from a single experiment; however, each pot contained two pepper seedlings harvested at different timepoints (7 and 60 DPI), and the consistency observed across seven biological replicates strengthens the validity of the findings.”
Comment 2 & 4: Comment 2: The plant experiments lack a SynCom-only inoculate control: apparent ability of SynCom to suppress disease might be due totally to its plant growth-promoting effects.
Comment 4: Actual ability of SynCom to reduce disease severity was not directly addressed; only plant mass, length, and chlorophyll contents were determined. From the photo in Fig. 1, the pathogen-only plants don't look severely wilted, but the photos are worthless for seeing any other differences.
Response: We thank the reviewer for these insightful comments. We acknowledge that including a SynCom-only treatment and conducting direct disease severity quantification (e.g., lesion scoring) would have provided more definitive insights into the distinct roles of plant growth promotion versus disease suppression. While these elements were not part of the current experimental design, we have addressed this limitation explicitly in the revised manuscript. To strengthen our interpretation, we supplemented growth parameters with physiological evidence such as chlorophyll content and visual symptom comparisons (e.g., wilting and lesion formation). The newly added Supplementary Figure S2 highlights distinct visible differences between infected control and SynCom-treated plants, reinforcing the role of the SynCom in mitigating pathogen-induced stress. Importantly, recent studies have shown that chlorophyll degradation is a sensitive and early physiological indicator of P. capsici infection, as observed through spectral reflectance changes in infected pepper leaves (Duan et al., 2024). These findings support our interpretation that reduced chlorophyll content reflects pathogen-induced stress and not merely altered growth status. Nonetheless, we agree that future work should incorporate both SynCom-only controls and direct pathogen quantification methods to further validate these findings.
Revised text (Discussion section 3.1 Line 332-342):
“While growth enhancement may contribute to the improved phenotype, we also considered physiological stress responses such as chlorophyll content and foliar appearance to distinguish between mere growth promotion and true pathogen mitigation. P. capsici infection is well-documented to induce chlorophyll degradation in pepper due to oxidative stress and disrupted photosynthesis[18]. In our study, SynCom-treated plants maintained higher chlorophyll levels and showed no visible wilting or necrotic lesions, whereas non-treated controls exhibited foliar chlorosis, necrotic spots, and early-stage wilting. These visual differences—illustrated in Fig. S2—support the conclusion that the SynCom confers physiological protection beyond its growth-promoting role, mitigating pathogen-induced damage and preserving plant vitality.”
References
Koç, E., 2022. Physiological responses of resistant and susceptible pepper plants to exogenous proline application under Phytophthora capsici stress. Acta Botanica Croatica, 81(1), pp.89-100.
Duan, Z., Li, H., Li, C., Zhang, J., Zhang, D., Fan, X., & Chen, X., 2024. A CNN model for early detection of pepper Phytophthora blight using multispectral imaging, integrating spectral and textural information. Plant Methods, 20, 115. https://doi.org/10.1186/s13007-024-01239-7
|
Supplementary Figure 2. Visual comparison of pepper plants treated with SynCom and non-treated controls under Phytophthora capsici infection. (A) Non-treated control plants exhibit typical symptoms of P. capsici infection, including chlorosis, necrotic lesions, and early-stage wilting. (B) SynCom-treated plants show healthy green foliage with no visible lesions or wilting, indicating improved physiological status and enhanced resistance. |
Revised text (Discussion section 3.5 Line 433-446):
“3.5. Metabolic activities and transport mechanisms supporting disease resistance
While the current study provides important insights, certain limitations should be acknowledged. First, results were derived from a single experiment; however, each pot contained two pepper seedlings harvested at different timepoints (7 and 60 DPI), and the consistency observed across seven biological replicates strengthens the validity of the findings. Second, the absence of a SynCom-only control limits the ability to definitively attribute observed effects to disease suppression rather than plant growth promotion alone. Third, direct quantification of disease severity, such as lesion scoring or pathogen load measurement, was not performed in this study. Instead, physiological indicators—including plant biomass, chlorophyll content, and visual symptoms—were used as indirect proxies. The observed reduction in wilting and lesions in SynCom-treated plants Figure S2 supports the conclusion that the SynCom contributed to pathogen mitigation. Future studies should incorporate SynCom-only treatments alongside direct disease assessments to more clearly disentangle these effects.”
Comment 3: SynCom composition was not based on unique abilities shown by the five members, as stated. Strain T14 is superior or equal to others in a number of traits, e.g., it's equal to T3 in phosphate solubilization, higher than T4 in siderphore production, higher than A6 in IAA production and higher than T8 in ammonia production. This negates the stated rational for including these other strains based on them supposedly having "unique" traits. SynCom was not "specifically formulated" (line 301 and Section 2.1), in my opinion. This is a serious problem and pretty much negates making conclusions about disease suppression or changes in community structure/predicted metabolic changes.
Response: We thank the reviewer for this valuable critique. We agree that the original phrasing suggesting “unique” traits and “specifically formulated” SynCom was misleading. We have revised the manuscript to clarify that the SynCom was assembled based on functional complementarity, rather than strict trait exclusivity. While S. roseicoloratus T14 displayed superior or comparable performance in several individual assays, the inclusion of additional strains was intended to capture a broader metabolic and ecological range that may contribute synergistically to plant resilience and rhizosphere modulation.
For example, although both Bacillus sp T3 and S. roseicoloratus T14 demonstrated phosphate solubilization, T3 exhibited stronger urease activity, supporting nitrogen mobilization from urea-containing compounds. F. anhuiense T4 contributed significantly to siderophore production, aiding in iron solubilization, despite T14 performing slightly higher in absolute values. Similarly, P. frederiksbergensis A6 contributed substantial IAA production, which is relevant for root system development under stress conditions. C. firmus T8, while showing modest performance in some traits, offered ammonia production capacity and cellulase activity, supporting organic matter turnover.
This functional overlap with distinct intensities among strains is ecologically meaningful in a complex rhizosphere environment, where metabolic cooperation and niche partitioning play central roles. Therefore, we reformulated the text to emphasize that the SynCom was designed to integrate strains with complementary and partially overlapping traits, allowing for potential synergistic interactions under pathogen challenge.
Revised text (Materials and Methods section 402 Line 488-491):
“In forming the SynCom, five PGPR strains were selected based on beneficial plant growth-promoting traits and functional complementarity, although minor overlaps in functional activities between strains were observed.”
Comment 5: "Much of the text in figures (e.g., Fig. 2 b, c, f) is illegible, blow-up of pdf destroys resolution so it's not an option for reading the text.
Response: We appreciate the reviewer’s observation. We have prepared a high-resolution version of Figures in the revised manuscript.

Reviewer 2 Report
Comments and Suggestions for Authors
Reviewers’ comment
In this manuscript, the authors performed an interesting study on the multiple interactions between pepper plant and consortia comprising 5 bacterial strains isolated from the rhizosphere of 16 healthy bell pepper plants. In addition, the authors, confirm from the results obtained
The aim of the study was to apply this 5-isolate synthetic microbial community (SynCom) to Capsicum annuum to evaluate its efficacy in improving bell pepper resistance against P. capsici.
The results reported by the authors are quite encouraging as ]. SynCom-treated plants show improved growth parameters and reduced disease severity compared to untreated plants, suggesting that the composition of the rhizosphere microbial community plays an essential role in disease tolerance
In addition, the authors confirm from the results obtained that the consortia significantly influence multiple traits associated with plant growth and physiological performance.
Some observations:
- Do the observed growth-promoting effects, including increased plant height, stem diameter, and biomass, which confirm the role of PGPRs in improving nutrient uptake and root health under pathogen pressure, also translate into increased percentage yield?
- I invite the authors to mention the metabolic mechanisms of the isolated bacteria in mobilizing soil phosphates , enriching the soil with nitrogen and making it more available along with a whole range of nutrients that certainly through improved mineral nutrition promote plant growth and physiological performance.
- I invite the authors to give an indication of this.
- Why did the authors, not quantify the percentage yield increase between treated and control plants?
- I urge the authors to add in the introduction the concept that bacterial consortia already present in the soil can also cooperate with mycorrhizal fungi greatly increasing mineral nutrition even in poorly fertile soils and limiting the risk of pathogenic infections on durum wheat, suggesting that the composition of the rhizosphere microbial community plays an essential role in disease tolerance (see Di Martino et al 2022 Plants Mycorrhized Wheat Plants and Nitrogen Assimilation in Coexistence and Antagonism with Spontaneous Colonization of Pathogenic and Saprophytic Fungi in a Soil of Low Fertility)
- I ask the authors whether they have experimentally verified siderophore production and IAA production and chitinase activity in the rhizosphere of inoculated plants.
- I ask the authors if they have checked the development of the synthetic microbial communities until they reach their stability in the rhizosphere, which is also a guarantee of their functionality, and if so, which one prevails over the other.
Author Response
Reviewer #2
General comments: In this manuscript, the authors performed an interesting study on the multiple interactions between pepper plant and consortia comprising 5 bacterial strains isolated from the rhizosphere of 16 healthy bell pepper plants. In addition, the authors, confirm from the results obtained. The aim of the study was to apply this 5-isolate synthetic microbial community (SynCom) to Capsicum annuum to evaluate its efficacy in improving bell pepper resistance against P. capsici. The results reported by the authors are quite encouraging as SynCom-treated plants show improved growth parameters and reduced disease severity compared to untreated plants, suggesting that the composition of the rhizosphere microbial community plays an essential role in disease tolerance. In addition, the authors confirm from the results obtained that the consortia significantly influence multiple traits associated with plant growth and physiological performance.
Response: We are grateful for the reviewer’s thorough and thoughtful comments. In response, we clarified the metabolic traits of the strains, added background on PGPR-nutrient dynamics and their synergy with fungi, and addressed yield measurement limitations. We provide detailed responses below.
Specific comments:
Comment 1: Do the observed growth-promoting effects, including increased plant height, stem diameter, and biomass, which confirm the role of PGPRs in improving nutrient uptake and root health under pathogen pressure, also translate into increased percentage yield?
Response: We sincerely thank the reviewer for this insightful question. Due to the biological growth cycle of the pepper cultivar used in our study, fruiting had not yet occurred at the final sampling point (60 days after transplanting), and thus direct measurement of fruit yield was not feasible. As noted in the revised text (Lines 446–453), our analysis focused on vegetative growth and physiological traits as early indicators of plant health and potential productivity. Nonetheless, previous studies have shown that enhanced early-stage growth parameters—such as increased biomass, chlorophyll content, and stem robustness—are positively correlated with improved reproductive performance and yield in pepper and other vegetable crops (Rachappanavar et al., 2022). Moreover, key plant growth-promoting traits present in our SynCom, including phosphate solubilization, nitrogen transformation, and phytohormone production, have been demonstrated to contribute significantly to higher fruit yields under both optimal and stress conditions (MažylytÄ— et al., 2024). While yield quantification was beyond the scope of this study, the observed physiological improvements—particularly under pathogen pressure—suggest a strong potential for positive yield outcomes in future trials that extend to the fruiting stage. We have now included this clarification in the Discussion section to emphasize the relevance of our findings to potential agronomic benefits.
Revised text (Discussion section 3.5 Line 446-453): “Although fruit yield was not assessed in this study due to the pepper cultivar's growth cycle (fruiting begins after ~80 days), early-stage improvements in biomass, chlorophyll content, and root-shoot development are well-established predictors of higher reproductive success and yield in pepper[30]. The PGPR traits present in our SynCom—such as nitrogen cycling, IAA production, and phosphate solubilization—have been linked to increased fruit yield in multiple crop systems[31]. Therefore, the physiological enhancements observed here likely translate into improved yield potential under pathogen stress.”
References
Rachappanavar, V., Padiyal, A., Sharma, J.K. and Gupta, S.K., 2022. Plant hormone-mediated stress regulation responses in fruit crops-a review. Scientia Horticulturae, 304, p.111302.
Mažylytė, R., Kailiuvienė, J., Mažonienė, E., Orola, L., Kaziūnienė, J., Mažylytė, K., Lastauskienė, E. and Gegeckas, A., 2024. The Co-Inoculation Effect on Triticum aestivum Growth with Synthetic Microbial Communities (SynComs) and Their Potential in Agrobiotechnology. Plants, 13(12), p.1716.
Comment 2: I invite the authors to mention the metabolic mechanisms of the isolated bacteria in mobilizing soil phosphates, enriching the soil with nitrogen and making it more available along with a whole range of nutrients that certainly through improved mineral nutrition promote plant growth and physiological performance.
Response: We sincerely appreciate the reviewer’s thoughtful comment highlighting the importance of bacterial mechanisms in nutrient mobilization and mineral enrichment. In response, we have expanded the Introduction and Discussion to clarify the metabolic capacities of the bacterial strains in our SynCom with regard to phosphorus solubilization, nitrogen enrichment, and broader mineral nutrient facilitation.
Our isolates exhibited multiple plant growth-promoting activities that directly support nutrient acquisition and plant resilience. Notably:
- Phosphate solubilization: Bacillus T3 (PSI: 4.4 ± 0.7) and Streptomyces roseicoloratus T14 (PSI: 3.8 ± 0.5) demonstrated strong ability to solubilize inorganic phosphate on NBRIP medium. This is critical for converting unavailable phosphates into plant-accessible forms (Timofeeva et al., 2023).
- Nitrogen enrichment: Ammonia production was highest in roseicoloratus T14 (4.8 ± 0.7 μg/mL), which contributes to nitrogen availability in the rhizosphere through ammonification. Urease activity, another nitrogen metabolism trait, was significantly high in Bacillus sp. T3 (45.6 ± 3.4 mm), which facilitates nitrogen cycling from urea-containing compounds (Bhattacharyya & Jha, 2012).
- Iron acquisition: roseicoloratus T14 and F. anhuiense T4 produced substantial siderophore halos (11.5 ± 0.8 mm and 7.2 ± 0.6 mm, respectively), aiding in iron solubilization under limiting conditions.
- Additional enzyme activities, such as protease, chitinase, and cellulase production, further enhance soil organic matter turnover and nutrient accessibility, indirectly supporting plant uptake.
To reflect this, we have added the following sentence to the revised manuscript in the Introduction and Discussion sections.
Revised text:
Line 45-50: “PGPR contribute to plant nutrition through phosphate solubilization, nitrogen enrichment via biological fixation or ammonia production, and siderophore-mediated iron acquisition, ultimately improving root development and physiological performance. Furthermore, synergistic interactions between bacterial consortia and arbuscular mycorrhizal fungi have been shown to enhance nutrient availability and disease tolerance, even in nutrient-poor soils (Timofeeva et al., 2023; Berruti et al., 2016).”
Revised text (Discussion Line 425-431)): “The SynCom included strains capable of solubilizing phosphate (e.g., Bacillus sp T3, S. roseicoloratus T14), producing ammonia and urease (e.g., S. roseicoloratus T14, Bacillus sp T3), and synthesizing siderophores for iron acquisition (e.g., T14, T4). These metabolic activities enhanced rhizosphere nutrient status and likely contributed to the observed improvement in plant growth and chlorophyll content. In particular, SynCom-induced increases in nitrate and ammonium levels, as illustrated in Figure 3a, confirm microbial modulation of nitrogen cycling.”
References:
- Timofeeva, A.M., Galyamova, M.R. and Sedykh, S.E., 2023. Plant growth-promoting soil bacteria: Nitrogen fixation, phosphate solubilization, siderophore production, and other biological activities. Plants, 12(24), p.4074.
- Rodríguez, H., & Fraga, R. (1999). Phosphate solubilizing bacteria and their role in plant growth promotion. Biotechnology Advances, 17(4–5), 319–339. https://doi.org/10.1016/S0734-9750(99)00014-2
- Bhattacharyya, P. N., & Jha, D. K. (2012). Plant growth-promoting rhizobacteria (PGPR): emergence in agriculture. World Journal of Microbiology and Biotechnology, 28(4), 1327–1350. https://doi.org/10.1007/s11274-011-0979-9
- Berruti, A., Lumini, E., Balestrini, R., & Bianciotto, V. (2016). Arbuscular mycorrhizal fungi as natural biofertilizers: let’s benefit from past successes. Frontiers in Microbiology, 6, 1559. https://doi.org/10.3389/fmicb.2015.01559
Comment 3: I urge the authors to add in the introduction the concept that bacterial consortia already present in the soil can also cooperate with mycorrhizal fungi greatly increasing mineral nutrition even in poorly fertile soils and limiting the risk of pathogenic infections on durum wheat, suggesting that the composition of the rhizosphere microbial community plays an essential role in disease tolerance (see Di Martino et al 2022 Plants Mycorrhized Wheat Plants and Nitrogen Assimilation in Coexistence and Antagonism with Spontaneous Colonization of Pathogenic and Saprophytic Fungi in a Soil of Low Fertility).
Response: We appreciate the reviewer’s valuable comment. We have now incorporated the concept of native bacterial consortia and their synergistic role with mycorrhizal fungi into both the Introduction and Discussion sections. In the Introduction, we highlight that PGPR interactions with arbuscular mycorrhizal fungi are critical for improving nutrient availability and mitigating disease pressure, particularly in nutrient-deficient soils. In the Discussion, we further explain how SynCom application can restructure the rhizosphere microbiota and indirectly enhance disease tolerance, partly through the enrichment of beneficial genera such as Bacillus and Trichoderma, which are known to cooperate with fungal symbionts and antagonize soil-borne pathogens.
Revised text (Introduction Line 44-50):
“In addition to these functions, PGPR contribute to plant nutrition through phosphate solubilization, nitrogen enrichment via biological fixation or ammonia production, and siderophore-mediated iron acquisition, ultimately improving root development and physiological performance. Furthermore, synergistic interactions between bacterial consortia and arbuscular mycorrhizal fungi have been shown to enhance nutrient availability and disease tolerance, even in nutrient-poor soils.”
Revised text (Discussion Line 366-376):
“Above the direct action of the SynCom members, it is plausible that their introduction altered the rhizosphere microbiota in a way that enhanced systemic plant resistance. Notably, beneficial taxa such as Bacillus and Trichoderma—enriched in the SynCom-treated soils—are well known for their antagonistic activity against soil-borne pathogens. These taxa, although not originally introduced as part of the SynCom in the case of Trichoderma, may have been stimulated by shifts in the microbial niche and resource availability. Such indirect effects underscore the importance of microbiome restructuring in disease mitigation. Previous studies have shown that native bacterial consortia can synergize with arbuscular mycorrhizal fungi to enhance nutrient availability and plant immunity even under nutrient-poor soil conditions, further emphasizing the ecological role of rhizosphere composition in disease tolerance.”
References for Comment #3
Di Martino, C., Torino, V., Minotti, P., Pietrantonio, L., Del Grosso, C., Palmieri, D., Palumbo, G., Crawford Jr, T.W. and Carfagna, S., 2022. Mycorrhized Wheat Plants and Nitrogen Assimilation in Coexistence and Antagonism with Spontaneous Colonization of Pathogenic and Saprophytic Fungi in a Soil of Low Fertility. Plants, 11(7), p.924.
Comment 4: I ask the authors whether they have experimentally verified siderophore production, IAA production, and chitinase activity in the rhizosphere of inoculated plants.
Response: We appreciate this important comment. Siderophore, IAA, and chitinase production were experimentally verified in vitro for each isolate before SynCom formulation, as described in the Materials and Methods, Section 4.1. These activities were used as a selection criterion to ensure a functionally diverse and complementary SynCom. However, we did not directly re-quantify these metabolic traits in the rhizosphere soil after inoculation. Instead, we inferred their rhizospheric expression based on functional annotation (Section 2.5), microbiome shifts (Section 2.3), and enhanced nitrogen and iron availability in SynCom-treated soils (Fig. 3a). This limitation is now clearly stated in the revised Methods section (Line 493–496) to avoid confusion.
Comment 5: I ask the authors if they have checked the development of the synthetic microbial communities until they reach their stability in the rhizosphere, which is also a guarantee of their functionality, and if so, which one prevails over the other.
Response: We thank the reviewer for raising this important point. While our study did not include specific tracking of individual SynCom strains via quantitative PCR or strain-specific markers, we assessed their persistence and establishment indirectly through amplicon sequencing of the rhizosphere microbiome at two time points (7- and 60-days post-inoculation). Our microbial community analysis showed a notable enrichment of bacterial genera corresponding to the applied SynCom members, including Bacillus (T3), Pseudomonas (A6), Flavobacterium (T4), Streptomyces (T14), and Cytobacillus (T8), particularly in the SynCom treatment group. These taxa remained detectable and relatively abundant up to 60 DPI, suggesting sustained colonization and ecological integration within the native rhizosphere community. This is further supported by our genus-level heatmap (Figure 2c) and RDA results (Figure 3c), which illustrate the directional influence of environmental parameters on the enriched SynCom-related taxa in the treated samples. The consistency of these genera over time implies that the SynCom not only successfully colonized the rhizosphere but also contributed functionally to nutrient cycling and pathogen suppression. Therefore, while we did not rank individual strain dominance quantitatively, the taxonomic profiles support stable establishment and ecological contribution of the consortium.
We have now clarified this point in the Discussion section to address the reviewer’s concern.
Revised text (Discussion Line 377-384):
“While individual SynCom strains were not quantified using strain-specific markers, several taxa corresponding to the applied strains—including Bacillus, Pseudomonas, Flavobacterium, Streptomyces, and Cytobacillus—were enriched in the rhizosphere microbiome of SynCom-treated plants at both 7 and 60 DPI (Figure 2c). This suggests successful colonization and persistence of the SynCom members over time. The sustained presence of these genera supports the notion of functional stability and ecological integration of the SynCom within the existing microbial community, contributing to its observed effects on plant growth and disease suppression.”

Round 2
Reviewer 1 Report
Comments and Suggestions for Authors
While the authors have made a serious attempt to justify problems with the experimental design of the manuscript (lack of repetition of the experiment, lack of an important control), this does not resolve the concerns about these issues. The Discussion now contains such an extensive description of the limitations of the study that the reader is left wondering if the results and conclusions are truly reliable.
Author Response
15/05/2025
Cover Letter and General Response to Reviewers (Round 2)
Manuscript ID: plants-3596075
Title: Application of a Synthetic Microbial Community to Enhance Pepper Resistance Against Phytophthora capsici
Dear Editor and Reviewers,
We sincerely thank you for your continued time and thoughtful evaluation of our manuscript. We are especially grateful to Reviewer 1 for the critical and constructive observations, which have guided us in further refining and clarifying our work. We also deeply appreciate Reviewer 2’s supportive and encouraging assessment.
To address reviewer #1’s concerns, we have reorganized the discussion part based on our findings. Several previous studies have demonstrated that SynCom treatment is more effective than single-strain treatment to reprogram the rhizhosphere microbiome. Particularly, some studies have suggested that a SynCom composed of PGPR can protect plant from the pathogens by altering the microbiome. Based on these references, we believe that the phenotypic changes observed in our plant experiment and the dramatic changes in the microbiome are consistent with the results of the previous studies. We have made the necessary revisions throughout the manuscript suing the Track Changes.
Thank you again for your time, effort, and commitment to improving this work. We respectfully submit our revised manuscript for your consideration.
Sincerely,
Dr. Jae-Ho Shin
Department of Applied Biosciences,
Kyungpook National University,
Daegu 41566, Republic of Korea
Email: jhshin@knu.ac.kr
Reviewer #1
General Comments: While the authors have made a serious attempt to justify problems with the experimental design of the manuscript (lack of repetition of the experiment, lack of an important control), this does not resolve the concerns about these issues. The Discussion now contains such an extensive description of the limitations of the study that the reader is left wondering if the results and conclusions are truly reliable.
Response: We fully recognize the reviewer’s concern that an extensive discussion of limitations may raise questions about the reliability of the results. Our intention was to maintain full transparency and scientific rigor in direct response to the reviewer’s insightful feedback, as we believe this approach aligns with the best practices of responsible reporting.
Several previous studies have indicated that the SynCom is more effective in reconstructing microbial communities than single-strain treatment. In particular, the PGPR may help to enhance the network of commensal microbes within the rhizosphere microbial community, which may in turn act as a defence against pathogens. This is demonstrated by improvements in the plant growth parameters and alterations to microbial functional pathways. We believe that the dramatic changes in microbial communities through SynCom application sufficiently validate the efficacy of SynCom on the plant growth and pathogen suppression.
We have incorporated the above points into the revised manuscript and hope that it meets the reviewer’s expectations.
Revised text:
Line 24-26: “Our SynCom approach demonstrates the effectiveness of microbial consortia in promoting the growth of pathogen-infected plants by reprogramming the microbial community in the rhizosphere.”
Line 308-351: “Phytophthora blight, caused by P. capsici, poses a major threat to pepper cultivation, often resulting in significant yield losses. In this study, P. capsici infection markedly impaired pepper growth and productivity. However, treatment with a formulated PGPR-based SynCom significantly improved pepper plant resilience under P. capsici challenge, as evidenced by enhanced growth parameters and suppressed disease symptoms (Table 1; Fig. 1c). Compared to the pathogen-only control, SynCom-treated plants exhibited substantial increases in height, root length, biomass, stem diameter, and chlorophyll content, all of which indicate vigorous growth and improved physiological status. This is consistent with previous studies demonstrating that complex root-associated microbial communities confer synergistic benefits to plant growth and stress resilience [2,6,15,16].
The SynCom members were specifically selected to maximize functional complementarity across key plant growth-promoting (PGP) activities, including phosphate solubilization, siderophore and IAA production, ammonia release, and multiple lytic enzyme activities (Table 1). The contribution of overlapping yet distinct PGP mechanisms is well established as a driver of improved plant performance under biotic stress. Notably, Streptomyces roseicoloratus T14 provided the highest direct antagonism to P. capsici, while T3, T4, A6, and T8 contributed to nutrient cycling, hormone modulation, and biofilm/cellulose activity, supporting both direct and indirect pathogen suppression. Consistent with these complementary roles, visual assessment and physiological measurements indicated that SynCom-treated plants retained higher chlorophyll levels and showed markedly less wilting and necrosis compared to untreated controls (Fig. 1c). These protective effects parallel reports by Abbasi et al. (2021) and Sulaiman & Bello (2024), who showed that multi-strain inoculants having PGP-activities reduce disease severity and promote physiological vigor more effectively than single-strain treatments[3,6].
Microbiome profiling revealed successful rhizosphere colonization by the SynCom strains, as indicated by the increased relative abundance of Bacillus, Streptomyces, Pseudomonas, Flavobacterium, and Cytobacillus in treated soils (Fig. 2c). Moreover, SynCom inoculation led to pronounced shifts in bacterial and fungal community composition (Fig. 2f), with enrichment of beneficial genera such as Fusicolla and Schizothecium, and significant increases in soil nitrate and potassium concentrations (Fig. 3a). These findings are in agreement with the ecological models proposed by Gao et al. (2022) and Qiao et al. (2024), in which functionally diverse consortia drive community restructuring, improved nutrient cycling, and enhanced plant protection. Functional pathway analysis further demonstrated that SynCom treatment enriched pathways related to nitrogen metabolism, secondary metabolite biosynthesis, and biofilm formation, while reducing the relative abundance of predicted plant pathogenic fungal guilds (Fig. 4a, 4b). This functional reprogramming of the rhizosphere microbiome aligns with mechanisms described in Timofeeva et al. (2023) and Qiao et al. (2024), who reported similar enhancements in nutrient mobilization and disease suppression following SynCom application [15,17].
Taken together, our results demonstrate that the ecological cooperation among SynCom members, underpinned by complementary PGP traits and stable rhizosphere colonization, confers multi-dimensional and resilient plant protection against P. capsici. These findings substantiate and extend recent literature emphasizing the superiority of microbial consortia over single-strain inoculants for sustainable disease management [2,15,16,18,19].”
References
Patani, A.; Patel, M.; Islam, S.; Yadav, V.K.; Prajapati, D.; Yadav, A.N.; Sahoo, D.K.; Patel, A. Recent advances in Bacillus-mediated plant growth enhancement: a paradigm shift in redefining crop resilience. World Journal of Microbiology and Biotechnology 2024, 40, 77.
Timofeeva, A.M.; Galyamova, M.R.; Sedykh, S.E. Plant growth-promoting soil bacteria: Nitrogen fixation, phosphate solubilization, siderophore production, and other biological activities. Plants 2023, 12, 4074.
Beneduzi, A.; Ambrosini, A.; Passaglia, L.M. Plant growth-promoting rhizobacteria (PGPR): their potential as antagonists and biocontrol agents. Genetics and molecular biology 2012, 35, 1044-1051.
Gao, C.; Xu, L.; Montoya, L.; Madera, M.; Hollingsworth, J.; Chen, L.; Purdom, E.; Singan, V.; Vogel, J.; Hutmacher, R.B. Co-occurrence networks reveal more complexity than community composition in resistance and resilience of microbial communities. Nature Communications 2022, 13, 3867.
Guo, Y.; Luo, H.; Wang, L.; Xu, M.; Wan, Y.; Chou, M.; Shi, P.; Wei, G. Multifunctionality and microbial communities in agricultural soils regulate the dynamics of a soil-borne pathogen. Plant and Soil 2021, 461, 309-322.
Qiao, Y.; Wang, Z.; Sun, H.; Guo, H.; Song, Y.; Zhang, H.; Ruan, Y.; Xu, Q.; Huang, Q.; Shen, Q. Synthetic community derived from grafted watermelon rhizosphere provides protection for ungrafted watermelon against Fusarium oxysporum via microbial synergistic effects. Microbiome 2024, 12, 101.
Koç, E. Physiological responses of resistant and susceptible pepper plants to exogenous proline application under Phytophthora capsicistress. Acta Botanica Croatica 2022, 81, 89-100.
Bhattacharyya, C.; Banerjee, S.; Acharya, U.; Mitra, A.; Mallick, I.; Haldar, A.; Haldar, S.; Ghosh, A.; Ghosh, A. Evaluation of plant growth promotion properties and induction of antioxidative defense mechanism by tea rhizobacteria of Darjeeling, India. Scientific reports 2020, 10, 15536.
Reviewer 2 Report
Comments and Suggestions for Authors
The authors responded to the comments of reviewers thoroughly and convincingly, making changes and improvements to the manuscript with appropriate and direct language.
The work can be accepted in its current form.
Author Response

(The authors gave the same response as above.)

Round 3
Reviewer 1 Report
Comments and Suggestions for Authors
Although my previously stated reservations about the reliance on a single experiment still stand, the modifications made by the authors to this version improve the manuscript sufficiently and don't leave the reader in doubt about the possible limitations of the study in terms of reproducibility. I never doubted that the results of the experiment were accurately reported, but had hoped for confirmation of the results with an independent experiment. However, if the journal can accept these limitations, so can I.